# "Let him die. He caused it": A qualitative study on cancer stigma in Tanzania

Judith M. Mwobobia[1]*, Brandon A. Knettel[1,2], Jennifer Headley[1], Elizabeth F. Msoka[3,4,5], Clotilda S. Tarimo[3], Victor Katiti[3,5], Erika Juhlin[6], Nosayaba Osazuwa-Peters[1,6,7]

**1** Duke Global Health Institute, Duke University, Durham, North Carolina, United States of America, **2** Duke University School of Nursing, Durham, North Carolina, United States of America, **3** Kilimanjaro Clinical Research Institute, Moshi, Tanzania, **4** Kilimanjaro Christian Medical Centre, Moshi, Tanzania, **5** Kilimanjaro Christian Medical University College, Moshi, Tanzania, **6** Department of Head and Neck Surgery, Duke University Medical Center, Durham, North Carolina, United States of America, **7** Department of Population Health Sciences, Duke University, Durham, North Carolina, United States of America

\* judith.mwobobia@duke.edu

**Data Availability Statement:** There are ethical constraints to sharing the data. We are unable to make these data available for public deposition because open data sharing was not specified in

## Abstract

Cancer stigma presents a critical barrier to care seeking, contributing to delayed presentation and poor cancer outcomes worldwide. The burden of cancer in Tanzania is on the rise, with cancer being the third-leading cause of death in the country. Despite rising incidence and poor outcomes of cancer, cancer-related stigma interventions have received low prioritization. There is a need for sound research that focuses on understanding attitudes driving stigma, its impact on care-seeking and treatment adherence, and intervention models to reduce stigma. We used a cross-sectional qualitative study design. We administered three open-ended qualitative questions to 140 adults newly diagnosed with cancer in Moshi, Tanzania. The questions explored common attitudes toward people with cancer, the perceived impact of cancer-related stigma on care engagement, and ideas for reducing cancer stigma. Patients were recruited during routine appointments at the Cancer Center at Kilimanjaro Christian Medical Center. Data were analyzed using a team-based, applied thematic approach and NVivo 12 software. All participants described stigma as a significant challenge for treatment and receiving support from their social networks. Perceptions of financial burden, misconceptions about cancer, such as the belief that it is contagious, and fear of death, were common attitudes driving cancer stigma. Most participants feared that symptoms would prevent them from being able to work and that the cost of cancer care would drive away loved ones. Stigma was not a ubiquitous response, as some participants reported increased care and social support from family members after a cancer diagnosis. Experiences of stigma contributed to feelings of shame, fear of burdening the family, reduced resources to access treatment, and disengagement from care. Common substitutes to medical therapies included religious interventions and traditional medicine, perceived as less expensive and less stigmatizing. Many participants felt they would benefit from improved financial support, professional counseling, and education for families and communities to reduce stigmatizing attitudes and enhance social support. There is a need for intervention studies focused on improving cancer literacy, community advocacy to reduce cancer stigma, and increasing emotional and practical support for people with

participant informed consent, nor in the approved ethical protocols for this study. We also have concerns about patient confidentiality in the sharing of detailed qualitative data, as the patient population at this cancer center is quite small. Data requests can be made to Duke University IRB which can be reached by phone number (919) 668-5111. For this study, please reference Protocol No. Pro00108097.

**Funding:** This project was conducted with pilot funding from the Duke Global Health Institute. The grant was co-led by Dr. Nosayaba Osazuwa-Peters and Dr. Brandon Knettel. There is no grant number associated with this award. Brandon Knettel is supported by a Career Development Award from the U.S. National Institute of Mental Health (K08 MH124459). Nosayaba Osazuwa-Peters is supported by a Career Development Award from the U.S. National Institute of Dental & Craniofacial Research (K01 DE030916). The funders had no role in study design, data collection and analysis, decision to publish, or preparation of the manuscript.

**Competing interests:** I have read the journal's policy, and the authors of this manuscript have the following competing interests: Dr. Knettel is a scientific advisor for Cancer Support Community. Nosayaba Osazuwa-Peters is a scientific advisor for Navigating Cancer. The other authors have no competing interests, financial or otherwise, to report regarding this research.

cancer and their families. There is also a clear need for policy efforts to make cancer care more affordable and accessible to reduce the financial burden on patients and families.

## 1. Introduction

Cancer is a rising crisis in Africa. Projections show that between the years 2020 and 2040, the continent will have the highest increase in cancer incidence and mortality of any region worldwide [1]. Each year, most cancer deaths are reported worldwide- about 70% occur in low- and middle-income countries [LMICs], which include many African countries [1]. In Sub-Saharan Africa, cancer ranks second among all causes of mortality after cardiovascular diseases [2], with death rates more than three times the global average for several common cancer types such as cervical and prostate cancer [3].

The contributing factors for the high burden of cancer in sub–Saharan Africa are complex and include demographic, epidemiologic, patient-level, social, and treatment-level factors [4]. At the demographic and epidemiologic levels, Africa is also experiencing an aging population because of rising life expectancy, contributing to the upsurges of diseases that are more prevalent later in life [5]. At the patient level, lack of knowledge of cancer, screening, and treatment, as well as fear and internalized stigma, are essential influences on symptom recognition and treatment-seeking [6]. High death rates in the region are also attributable to treatment-level factors, including lack of facilities, delays in diagnosis and initiation of treatment, and limited access to advanced therapies, which lead to poor prognosis [7, 8]. Broader socioeconomic challenges may contribute to care affordability, infrastructure, and transportation [9]. Other social factors come into play. Some of the most frequently diagnosed cancers in Africa are associated with infections [e.g., HPV as a cause of cervical cancer], and the region has a low number of cancer control programs [4]. Moreover, although global trends indicate a decline in the overall number of individuals living in extreme poverty, it is crucial to recognize that the absolute number of people experiencing extreme poverty in Africa is, in fact, on the rise [10].

Despite rising incidence and poor outcomes of cancer across many African settings, the mental health aspects of cancer have received low prioritization, and there are few studies on cancer-related stigma with a focus on Africa [2, 11, 12]. To understand stigma better, this study uses the Weiss and Ramakrishna definition of stigma as "a social process or related personal experience characterized by exclusion, rejection, blame, or devaluation that results from experience or reasonable anticipation of an adverse social judgment about a person or group identified with a particular health problem" [13]. A recent study [14] done in Tanzania found four categories of cancer stigmatizing attitudes: enacted stigma, shame and blame, internalized stigma, and disclosure concerns. These generally mirror Earnshaw and Chadoir's HIV stigma framework [internalized, enacted, and anticipated] and previous adaptations of mental health and HIV stigma to cancer populations, suggesting some overlap between health stigmas in this context [15].

Internalized or self-stigma includes negative attitudes and beliefs and can lead to low self-esteem and other negative consequences. Enacted stigma involves actual mistreatment and discrimination from others. Anticipated or community stigma is the anticipation of stigma that may occur if one's identity becomes more widely known. Cancer stigma in each of these forms is a critical barrier to care-seeking, contributing to delayed presentation and poor cancer outcomes [15]. Previous research has shown associations between cancer-related stigma and poor mental health, leading to an increased likelihood of experiencing depression and anxiety [16].

There is, however, limited research that attempts to describe how cancer stigma is expressed and how it hinders care-seeking. No stigma-reduction interventions are designed specifically for low-resource settings like Moshi Tanzania. Stigma leads to poor self-esteem and isolation, which hinder patients from receiving vital social support [17]. Taken together, these stigma-related challenges can lead to poor utilization of cancer screening services, increased fear, delayed treatment-seeking, and lower treatment adherence [18, 19].

Further research is necessary to understand patient experiences of cancer stigma in low-resource settings in Africa. These studies can inform the development of psychosocial interventions that reduce the burden of cancer-related stigma, increase awareness of cancer, and improve adherence to cancer screening and treatment [20].

This study investigates cancer-related stigma and discrimination among individuals in Moshi, Tanzania. It identifies various forms of stigma experienced by individuals with cancer, including internalized, enacted, and anticipated stigma. We aim to highlight the complex interplay between societal perceptions and individual experiences of cancer-related stigma. The study seeks to inform practical interventions that address the unique challenges faced by individuals with cancer in low-resource settings by identifying specific beliefs contributing to stigma and exploring strategies to address them. We examine the beliefs and attitudes held about cancer and those affected by cancer. We also explore how cancer-related stigma influences support and care-seeking among these patients and the potential psychosocial interventions to reduce cancer stigma in this setting. The findings from this study can guide the development of stigma-reduction interventions tailored to low-resource settings.

## 2. Materials and methods

Our philosophical basis is rooted in the phenomenological philosophy approach [21]. Phenomenology aligns with our aim to understand the subjective realities of people as they navigate the social and cultural dimensions of cancer stigma within their lived context. Our methodological choices included using semi-structured interviews to gather rich, detailed participant accounts. Through a phenomenological philosophy lens, we aimed to capture the essence of the participants' lived experiences and uncover the complexities of cancer-related stigma.

We employed a thematic analysis qualitative research design approach by identifying predetermined areas of exploration and inductively coding qualitative data within them as we reviewed the information. The study examined beliefs about cancer, the impacts of cancer stigma, and potential areas of intervention. The qualitative data were collected as part of a more extensive survey administered to people newly diagnosed with cancer at a single cancer center in the Kilimanjaro region of Tanzania from January to July 2022. As part of the survey, participants were asked three semi-structured questions focused on different aspects of cancer-related stigma used in this qualitative analysis: 1. In Tanzania, how do people talk about or behave around cancer patients? 2. Does stigma influence the way people seek support or treatment related to cancer in Tanzania? In what ways? 3. What are some ways we can help people with cancer to overcome or cope with stigma? The research team intuitively developed the questions, including mental health and cancer care experts, in collaboration with Tanzanian cancer nurses and treatment providers.

The questions were first asked verbatim, but the research assistant was encouraged to use follow-up probes or prompts to encourage further sharing when appropriate. Because of the highly sensitive nature of the subject, we phrased the questions in a non-personal/indirect way. We believed participants would be more comfortable describing the experiences of people with cancer in general terms, and this phrasing also empowered participants to define their

boundaries when deciding how much of their personal experience they would invoke in discussing the topic.

Qualitative research is a robust approach for generating new knowledge or assessing the fit of existing knowledge to new contexts. In this case, we wanted to understand the nature, impact, and scope of stigma from the experiences of cancer patients in Tanzania, where few studies on this topic have been conducted. We also wanted to identify the potential areas of intervention and possible solutions to the stigma. The data are intended to inform the design of new mental health or social support interventions to reduce cancer-related stigma and mitigate its negative consequences for care engagement.

## 2.1 Ethics statement

Relevant approvals to conduct this research were granted by the institutional review boards of the Duke University Health System (Protocol ID: Pro00108097), Kilimanjaro Christian Medical Centre (Protocol ID: 1307), and the National Institute of Medical Research in Tanzania (Protocol ID: 2122). All the participants provided informed written consent.

## 2.2 Setting

The study was conducted at the Cancer Care Centre at Kilimanjaro Christian Medical Centre (KCMC) in Moshi, Tanzania. KCMC is one of four zonal referral hospitals in Tanzania, operated by the Tanzanian Ministry of Health, and essentially serves patients from the seven administrative districts of the Kilimanjaro region, which has a population of approximately 1.8 million people. KCMC is among the largest hospitals in Tanzania, with about 650 beds, and the hospital contains many specialty services, such as the cancer center, a burn unit, ophthalmology, and psychiatry, available at only a handful of hospitals in the country.

## 2.3 Participants

The qualitative data collection was part of a more extensive structured survey of 154 participants newly diagnosed with cancer at the KCMC Cancer Centre. Patients were recruited from inpatient and outpatient clinics at the Centre. Patients were eligible for the parent study if they had received a diagnosis with any cancer type within a window of 1 to 6 months, were ages 18 and above, could communicate in either Swahili or English and were physically and cognitively capable of giving informed consent and completing the study survey. We waited one month to survey a new cancer diagnosis to allow the participant time to process the diagnosis and seek emotional support. The assessment of participants' physical and mental capacity for the study was done by the clinic nurses and the research assistants (RAs). Bedridden, critically ill, and patients who would experience challenges in talking for prolonged periods were ineligible for the study to avoid undue burden on these individuals.

## 2.4 Procedures

Using convenience sampling, participants were recruited for the study during routine cancer care at the Cancer Care Centre. Clinical nurses would first identify eligible participants at clinic check-in, provide them with basic information about the study, and refer those interested to a private office at the clinic to meet with a research assistant. The RAs (CT, EM, VK) confirm patient eligibility, read the informed consent form aloud, answer any questions, and then have the participants sign and keep a copy. Participants unable to write provided a thumbprint. The research assistant would offer the participant snacks, log into the Research Electronic Data Capture application (REDCap) [22, 23] on a study-owned tablet and begin

administering the survey. All questions were read aloud to the participants, and responses were recorded and entered directly into REDCap.

Participants' responses to the three open-ended questions were communicated in Kiswahili or English based on the participants' preferences. Responses were transcribed verbatim on a paper form. Names and identifying information were not recorded. Later, the same day as the survey was administered, the research assistant translated Kiswahili responses into English and entered them into REDCap. Once all data entry was completed, the research assistant transferred the records from the application to a secure online database hosted at Duke University.

The full survey took approximately 45 minutes to complete. The three open-ended questions, which are the focus of this study, were asked after the survey and took about 5 minutes to complete. After the survey, participants received 5,000 Tanzanian shillings (equivalent to USD 2.25) to compensate them for their time.

## 2.5 Data quality assurance

All data collection for our study on cancer stigma in Moshi, Tanzania, was conducted meticulously and adhered to clearly defined protocols to maintain high-quality data. The data were collected by a team of three-CT, EM, and VK, who are bachelor's-trained RAs with prior data collection experience. These RAs underwent initial training provided by the study's co-principal investigator (BK), a mental health provider with expertise in qualitative research and extensive experience in Tanzania. Refresher training sessions were organized for the RAs as needed. A second bilingual research team member (JM) reviewed translated responses to ensure quality, accuracy, and linguistic and cultural equivalence during the analysis. All the collected data were securely stored in REDCap, and access protocols were instituted to safeguard against unauthorized access or corruption.

Triangulation methods, such as having multiple researchers - JM, BK, EJ, NP, and JH carry out analysis, were utilized to enhance the credibility of the qualitative study. The diversity of perspectives and expertise among the researchers ensured a more nuanced understanding of the data, minimizing the impact of individual biases and contributing to the credibility of the results. The research team also acknowledged their positions and potential biases during data analysis to ensure a reflexive approach. In weekly meetings, the team discussed analytical code-based memos. Patterns and salient themes across the data were captured. Saturation was assessed by monitoring themes' recurrence and new insights' emergence throughout the analysis. The team thoroughly examined transferability, considering the characteristics of the sample and acknowledging any limitations in generalizing findings to other contexts.

Lastly, peer debriefing was employed as a strategy for external validation, engaging colleagues from the cancer center and hospital staff who were not directly involved in the research. These external colleagues provided feedback during debriefing sessions, contributing valuable insights to refine our interpretations and ensure methodological rigor. Future research recommendations include exploring identified themes in greater depth and investigating specific subgroups within the diverse participant pool.

## 2.6 Analysis

The more extensive study enrolled 154 participants; however, 14 were excluded from the qualitative analysis due to incomplete survey responses. Open-ended questions were administered after the more extensive survey, and a subset of participants, unable to complete these due to time constraints, were consequently excluded. The remaining 140 participants, characterized

by diverse demographics, including age, socioeconomic status, and cancer type, were included in the qualitative analysis.

NVivo 12 was utilized for the applied thematic analysis [24], organization, and analysis of de-identified open-ended survey data, allowing for a systematic and efficient exploration of patterns and themes. To foster transparency and reliability, 28 sets of responses (20%) were randomly selected for re-coding by a second reviewer, achieving an inter-coder agreement of 83.2%. The pre-established threshold was 80% agreement [25]. Any disagreements identified during the re-coding process were thoroughly discussed and reconciled by the two reviewers until a consensus was reached. Throughout the study, the entire research team convened weekly to discuss progress, address recruitment and data collection challenges, and refine the data analysis process. A coordinated team approach was employed to ensure the reliability and validity of our qualitative data analysis. The team conducted weekly data audits to continually monitor and uphold data quality, ensuring that the data consistently met high-quality standards throughout the study period

## 3. Results

The sample for this study consisted of 140 participants, with a mean age of 56 years, a median of 58 years, and a range of ages between 18 and 93. Most participants were female, accounting for 58% (n = 82) of the sample, while 42% (n = 58)were male. As illustrated in Fig 1, the most significant proportion of participants had Stage 4 cancer (38%), while only 6% of the study sample had Stage 1 cancer, representing high severity in this sample.

The most common type of cancer among female participants was breast cancer, while the most common type of cancer among male participants was prostate cancer.

Four qualitative domains were identified. These were beliefs contributing to cancer stigma, behaviors and attitudes toward people with cancer, the impact of stigma on patients, and strategies to reduce cancer stigma. Fig 2 shows how beliefs impact behavior, leading to consequences of stigma. A summary of the significant identified themes coded onto each of these domains can be found in Table 1.

### 3.1 Beliefs contributing to cancer stigma

The study participants were asked how individuals in Tanzania perceive those affected by cancer. The responses revealed prevalent beliefs held within the community regarding cancer. These beliefs were shared by both people with and without cancer. Some common beliefs included the mistaken idea that cancer is contagious, the perception that it is a death sentence, and the belief that it is expensive to treat. There was also a general perception of cancer as a mysterious illness. The data didn't show any distinct patterns or indications of connections between cancer type, severity, and the themes associated with beliefs about cancer.

**3.1.1 Cancer as a death sentence.**   56% of participants believed that cancer was a death sentence, with women more likely (68%, n = 48) than men (32%, n = 30) to hold this belief. Only 10.0% (n = 14) believed that cancer is treatable. This belief contributes to stigma among cancer patients.

A 74-year-old female participant with ovarian cancer said that beliefs about cancer being untreatable cause hopelessness:

> *"Cancer is a very dangerous disease which ends up in death. When people know one is a cancer patient, they lose hope in them and they say the person has no life anymore. They keep distance from the patient because they can't make it anymore."*

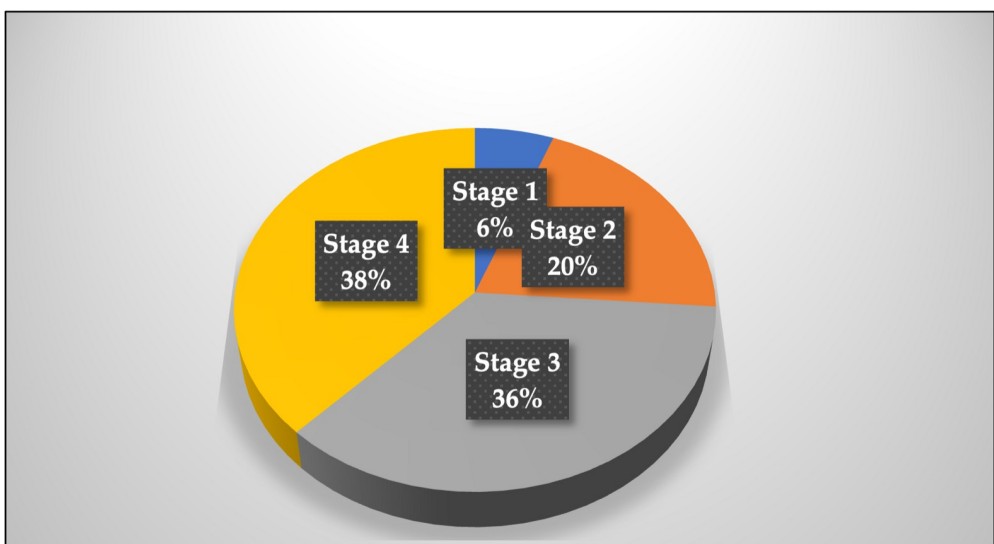

**Fig 1. Cancer stages among participants.**

A 62-year-old female multiple myeloma patient shared that a lack of knowledge about cancer can contribute to the belief that cancer is a death sentence. She said:

> *"I never knew that there was a treatment that could cure cancer. I thought if I was found to have cancer, I would die. Cancer is thought to be a threatening disease. Once you have it, you will never finish a year. I was so scared."*

**3.1.2 Beliefs around causes of cancer.** Over one-third of participants (36%, n = 50) reported that certain beliefs about the causes of cancer contributed to the stigmatization of cancer patients. These beliefs included misconceptions that cancer is sexually transmitted, caused by poor nutrition, sexual activity, ulcers, or that breast cancer was caused by a failure to breastfeed. A 55-year-old female breast cancer patient shared that when people learn about a cancer diagnosis, they blame the patient: *"They say, let him die with cancer. He caused it on his own."*

A 64-year-old male participant with prostate cancer shared that in his community, promiscuity was believed to be a cause of some cancers:

> *"They say if a patient has cervical cancer, it means she was having sex with many men."*

Some participants also reported that many people in their communities believe that God caused cancer, and a few participants, like a 75-year-old male prostate cancer patient, shared that some people believe it was caused by evil supernatural forces.

> *"People say the disease is brought by God so that we can turn to him. Others think cancer is associated with witches."*

These beliefs can lead to stigmatization and discrimination against cancer patients and can make it difficult for them to seek treatment.

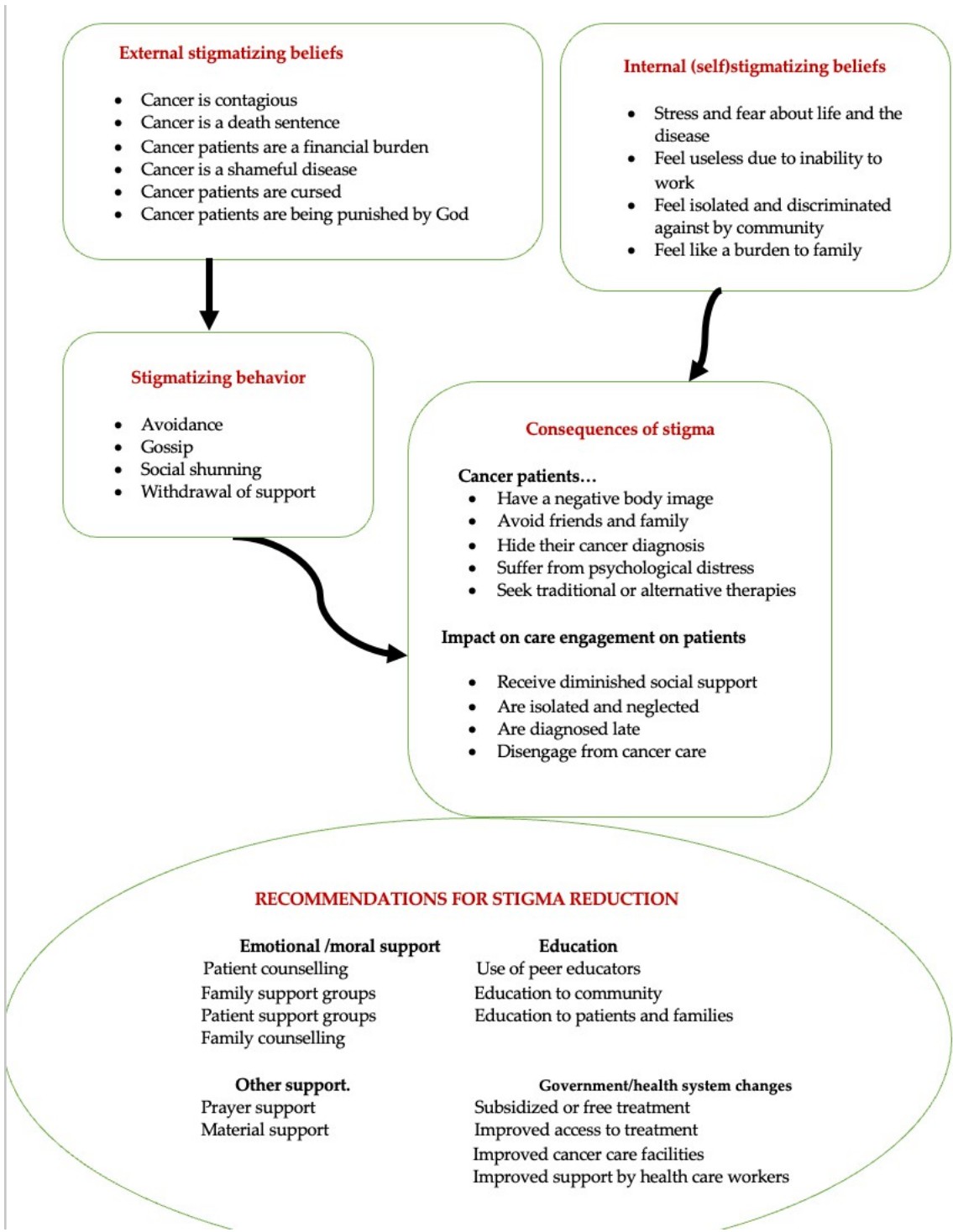

**Fig 2. Conceptual framework of cancer stigma derived from the qualitative themes.**

**3.1.3 Cancer is contagious.** The study found that 15% of participants (n = 21) believed cancer is contagious, contributing to stigma. This belief led to isolation of cancer patients by friends and family, fostering self-stigma. Some patients withheld their diagnosis to avoid

**Table 1. Summary of qualitative themes related to cancer stigma.**

| Theme | Frequency | Illustrative Quotation |
|---|---|---|
| **Beliefs Contributing to Cancer Stigma** | | |
| Death sentence, associated with death | 78 | "Having cancer is the end of life. When you pass by people who know you have cancer, they say that you will soon be dead." |
| Expensive to treat | 47 | "Patients get no support because cancer causes poverty in the family, so family members isolate the patient and then patients stop seeking treatment." |
| Causes of Cancer | 34 | "People say that cancer is a witchcraft disease that is sent to you by bad people." |
| Contagious | 21 | "They take a cancer patient like a person with HIV; if you sit close to a cancer patient, they can transmit it to you." |
| It is treatable | 14 | "Cancer is a disease like any other. If a cancer patient starts treatment early, they get cured. I have seen a woman who has been treated get cured." |
| Mysterious disease and general poor knowledge on cancer | 13 | "Many people still don't know what cancer is, especially in my village." |
| **Behaviors and Attitudes Toward People with Cancer** | | |
| Social isolation and discrimination, neglect | 56 | "We don't feel good sharing about the disease with friends since you don't know how they will react. I am an engineer. If people know I have cancer then they will feel that I can't work for them anymore." |
| Offer support | 37 | "Family members take care of cancer patient and assist them. Friends and neighbors visit patients for prayers and comfort." |
| Fear | 26 | "Cancer is a death sentence; it is a threatening disease. People fear the disease. I fear it too." |
| Hopelessness | 11 | "When a person is known to have cancer, people around him start to lose hope for him and stay away from him. They know he will not survive, so they keep a distance." |
| **Impact of Stigma on Patients** | | |
| Stigma leads to care disengagement | 68 | "A stigmatized patient may refuse treatment and feel as though they are no longer important to the family or community." |
| Stigma leads to people seeking traditional or alternative treatment | 39 | "Families tend not to take their patients to hospital. They think she will die sooner if they do that and instead take them to get traditional medicine because then she will live longer." |
| Diminished social support, neglect | 27 | "I don't get any support from my husband and relatives. So, I am now also a father and a mother to my children." |
| Psychological, emotional distress | 26 | "Some cancer patients get headaches from stress, because of fear of losing their partners. For example, since my husband left me, I started getting sick and now I am getting headaches more often." |
| No stigma or no effect | 16 | "There is no effect related to stigma on the way people seek support or treatment. I have not noted it. Stigma is more on HIV patients but not for cancer patients." |
| Negative body image and self-hate. | 5 | "Skin cancer makes a patient hate herself and hide from seeking treatment because of fear." |
| **Strategies to Address Stigma** | | |
| Education | 107 | "Education is needed in the community about cancer. They need to know that cancer is a disease like any other disease. It is not infectious, and it can be cured." |
| Government support & health system improvements | 42 | "The government should monitor the traditional healers on what they are doing and restrict them on dealing with serious issues like cancer." |
| Build hope, remove fear | 32 | "Counselling and comfort are needed by cancer patients. Counseling would make me strong and confident to continue with treatment." |

isolation. A 62-year-old female participant with multiple myeloma shared that cancer patients can develop mental distress due to isolation brought about by this belief. She said:

> *"Some people get stress due to the disease. Other families isolate them, thinking they will get cancer. I know people who think cancer is contagious. My family doesn't know I have cancer."*

**3.1.4 Cancer treatment is impoverishing.** Over a third of participants (34%, n = 48) noted that the high cost of cancer treatment contributes to stigma, causing distress. A 68-year-old male participant with prostate cancer emphasized the prolonged and costly nature of treatment, highlighting the need for external support. He said:

> *"The disease requires a lot of money, so if you are not financially well-off, you will need support. If you cannot be helped throughout, you will start to stigmatize yourself because you feel shy to beg people for [financial] support all the time."*

While undergoing treatment, patients often develop self-stigma or face stigma from the community because they may have to reduce their workload or stop working. Work challenges influence their ability to pay for their treatment and to support their families more generally.

A 44-year-old male participant with prostate cancer revealed that once family members realized the costs of treatment, they began to avoid him. He said:

> *"Relatives gave up because of treatment costs. Treatment costs are very high which leads people who are taking care of patients living with cancer to run away or isolate them."*

Other participants reported that a cancer diagnosis results in cancer patients being treated like a burden due to the associated high cost of treatment and the perception that they would become dependent on family members for care.

## 3.2 Behaviors and attitudes toward people with cancer

The most common forms of enacted stigma reported by people with cancer are social isolation and discrimination, which are often driven by feelings of fear and hopelessness. In some instances, cancer led to positive responses and increased support.

**3.2.1 Social isolation and discrimination.** Social isolation and discrimination emerged as prominent behavioral impacts of cancer stigma. Over a third of participants (41%, n = 57) reported negative changes in behaviors and attitudes from others following a cancer diagnosis. This manifested in exclusion from social activities or active avoidance. Notably, more female participants (66%, n = 37) than males (33%, n = 20) reported experiences of social isolation, discrimination, or both. Misconceptions and beliefs about cancer drive isolation and discrimination, as shared by a 61-year-old female colon cancer patient:

> *"Cancer patients sometimes are isolated due to the reason that they will die. Some family members get tired and start to mistreat patients by isolating them."*

Many participants revealed that cancer patients often avoid disclosing their diagnosis to prevent shunning or other forms of stigma. A 68-year-old male prostate cancer patient highlighted the potential discrimination faced in the workplace:

*"We don't feel good sharing about the disease with friends since you don't know how they will react. If people know I have cancer, then they will feel that I can't work for them anymore."*

While limiting disclosure may prevent discrimination, it can lead to the isolation of cancer patients who miss out on social support. An 18-year-old female Kaposi Sarcoma patient noted that limiting disclosure helps avoid humiliating situations:

*"[Cancer patients] fail to go and ask for help from their neighbors because they don't want to be embarrassed or be gossiped about."*

Discrimination extends to day-to-day activities, such as sharing meals. Some participants mentioned instances of refusing to share utensils or meals with cancer patients, expressing discomfort.

**3.2.2 Increased support from others.** About one-fourth of the participants (26%, n = 37) shared that they believed that once people found out that someone in their family or community was facing cancer, they rallied together and offered more support to them. This sentiment was shared equally by the male and female participants. Some participants attributed this response to religion. For example, a 65-year-old female breast cancer patient said:

*"In my village, if a patient is sick in bed, we do help them; we visit them in their homes, buy them food, wash, pray for them, and care for their families.*

Participants reported that support most often takes the form of encouragement, care, love, prayer, and company for the patient. A few participants mentioned financial support and practical support, such as accompanying the patient to the hospital, providing childcare, bringing food, and ensuring that the patient gets to the hospital as needed.

**3.2.3 Fear.** Many instances shared by participants pointed to the fear held by cancer patients as having the potential to develop into self-stigma. In a similar vein, participants often shared cases where fear of cancer in the community drove enacted stigma. Fear was associated with a cancer diagnosis by 18% (n = 25) of the participants, with similar responses among men and women. Interestingly, over half of the female participants who talked about fear were breast cancer patients. This fear, like a 69-year-old breast cancer patient says, can prevent one from seeking treatment.

*"The effects of chemotherapy may scare people. They say if I'm meant to die let me die like this."*

Besides breast cancer patients, participants with different cancers shared feelings of apprehension or fear around treatment procedures and effects. For example, a 74-year-old male suffering from urological cancer shared:

*"Cancer among men, prostate cancer, is a disease which undresses men for who they are. The examination is too painful and shameful because young doctors are examining us. When I was asked to complete the examination, I was about to say 'No' but my wife encouraged me."*

Participants also shared that worry and fear about how the family would react to the news about their diagnosis. A few participants, such as a 61-year-old female colon cancer patient, shared that sometimes fear was brought on by a lack of resources to tackle the challenges that lay ahead.

*"Cancer patients can stop their treatments, and sometimes they can stop attending clinic, because of fear to seek support such as money for transport or money for their medicine,"* she said.

A few participants reported that fear was also expressed by the community due to misconceptions around cancer, like 'cancer is contagious.' A 60-year-old male bladder cancer patient shared:

*"People fear cancer patients because they think cancer is a contagious disease. They don't even share activities they used to do before, like eating, drinking, or smoking cigarettes. They also don't go to their house anymore. They say it is an incurable disease and do not want to get it."*

### 3.3 Impact of stigma on patients

The third key domain was the influence of stigma on patients, which most often manifested in challenges with seeking care and treatment for cancer. The major themes identified in this domain included disengagement from care, seeking care from alternative and traditional healers, diminished social support and neglect, and psychological and emotional distress. A minor theme emerged in which participants expressed negative body image and self-hate, reflecting their unfavorable perceptions of their bodies. Notably, these responses were gathered from participants with various types of cancer. A smaller number of participants stated that stigma did not harm patients.

**3.3.1 Disengagement from care.**   Disengagement from care was the most mentioned consequence of stigma. About half of all participants (49%, n = 69) said that stigma caused cancer patients to never seek treatment or to stop seeking treatment after a diagnosis or initiation of care. An 18-year-old female participant with Kaposi Sarcoma shared that stigma brought on by fear or shame discouraged cancer patients from visiting hospitals.

*"They [patients] stop medications and decide to stay home because they feel bad, which leads to severe disease and even death due to stigma."*

Fear of being seen by friends or family, said a 68-year-old male participant with prostate cancer, makes patients actively avoid health centers. "Patients don't attend hospital because they feel shy to face other people they know in hospitals."

**3.3.2 Seeking alternative traditional or religious healers.**   Over one-quarter of participants (28%, n = 39) reported that stigma drove cancer patients to seek alternative treatments like herbalists, traditional healers, and religious interventions such as faith healing. Based on participants' responses, the widespread use of traditional healers and other alternative treatments led patients with suspicious and cancer-identifying symptoms to delay seeking screening or medical attention from hospitals. A 39-year-old female breast cancer patient said traditional healers are sought because their treatment is said to be more effective than conventional therapies.

*"Families tend not to take their patients to the hospital. They think she will die sooner if they do that and instead take them to get traditional medicine because then she will live longer."*

Other participants felt that traditional healers were chosen because they are less expensive than Western medicine. Some participants said that traditional healers were preferred because they provided more privacy to cancer patients and, in the face of stigma from the community, offered a safer space. A 68-year-old participant with prostate cancer shared:

*"The disease needs a lot of money, so. . .you will start to lose clinic attendance, or you will look for cheap medication like herbalists where you cannot be seen by many people."*

Another participant, a 71-year-old prostate cancer patient shared:

*"Stigma is caused by people's attitude towards patients. If a patient is not well counseled, stigma may make the patient seek treatment from an herbalist because the privacy there is greater."*

A 43-year-old male participant with hematological cancer shared that the media encourages cancer patients to seek alternative treatments.

*"Many advertisements talk about traditional healers being effective, and that is why many may delay getting help [from the hospital]."*

**3.3.3 Psychological distress.** About one-fifth of the participants (18%, n = 25) reported that stigma caused cancer patients to develop psychological distress. This theme was often linked to the previously mentioned theme of the high cost of treatment as a contributor to stigma. Financial difficulties, say some participants, contribute to psychological distress. A 57-year-old female participant with breast cancer shared that worry about finances can cause depression.

*"Lacking financial support due to illness and inability to work and inability to care for oneself makes a patient depressed and even leads to death."*

Most of the participants who mentioned this theme reported self-stigma as a key contributor to mental health challenges due to changes in one's physical appearance and body image.
A 65-year-old female breast cancer patient said far more than physical bodily changes cause distress.

*"Cancer makes patients feel bad about themselves and so they tend not to be seen by their old friends because cancer makes them weak and tired."*

Mental distress is also caused by fear of being isolated, said a 27-year-old female participant with Choriocarcinoma.

*"Cancer patients may get headaches and stress because of fear of losing their partners. For example, my husband left me when I started becoming sick and now, I am getting headaches more often."*

**3.3.4 No or minimal impact.** Nearly one in eight participants (12%, n = 17) reported that cancer stigma didn't influence how patients sought care or treatment. For example, a 25-year-old male participant with colorectal cancer said:

*"Stigma? No. It will not influence because we, the Maasai, when a person is sick, people come together to solve the problem."*

### 3.4 Strategies to support people with cancer

Three themes were identified under the domain describing strategies to improve support for people with cancer: education, government support, and building hope among cancer patients.

**3.4.1 Education.**  76% of participants (n = 107) believed education to be the best way to combat cancer stigma. Lack of knowledge about cancer and treatment options contributed to misconceptions and stigma. A 65-year-old female participant with breast cancer said, *"Many people still don't know what this cancer is, especially in my village."* The lack of knowledge, said a 41-year-old female participant with advanced breast cancer, contributed to late-stage presentation:

> *"Many cancer patients don't know if they have cancer and so they come to be diagnosed at late stage 3 or 4 like myself. I never knew I had cancer until it was too late."*

An 86-year-old male prostate cancer patient shared that the use of media would be one way to get the message across.

> *"Advertisements about cancers like prostate cancer. Many people have cancer, but they don't know. They should also be informed where to get such services, routine health checkups, to detect the disease early,"* he said.

26% of participants (n = 37) suggested education tailored to the broader community, while only 12% (n = 17) mentioned patient-focused education. Many participants shared that awareness of cancer stigma and how to cope with it were important aspects of patient-focused education and would bolster treatment adherence.

A 64-year-old male stomach cancer patient said:

> *"Cancer survivors can be good to offer education because I had a person who had cancer and he told me that cancer cannot be treated using local herbs. That is why I decided to come to the hospital."*

**3.4.2 Government support.**  Around 29% of participants (n = 41) suggested that the government should help reduce the stigma surrounding cancer in Tanzania. They recommended improving the public health system, subsidizing cancer treatment, and providing better screening and treatment access. Establishing new treatment facilities in underserved areas could reduce the financial burden and improve the quality of care. Such efforts could improve cancer outcomes, change the perception of cancer as a death sentence, and reduce associated stigma.

> *"Bring health care facilities close to where cancer patients are; this will help them to access treatment without running around begging for money for transport to go to hospital,"* said a 64-year-old female with breast cancer.

Another suggestion by participants was that support from health workers would help reduce stigma in the community. This could include community-based models of care, where health education, screening, and basic treatment are offered in community settings outside of traditional healthcare facilities.

*"Healthcare providers should pay a visit to churches and meetings and talk about cancer, stigma, and nutrition to help people understand it and prevent patients from experiencing stigma,"* said a 52-year-old male participant with esophageal cancer.

## 4. Discussion

Our study delved into the complexities of cancer stigma in Moshi, Tanzania. Grounded in Earnshaw and Chadoir's framework(14), we examined internalized, enacted, and anticipated stigma to unravel the intricacies of cancer-related social stigma. We acknowledged shared elements and distinct characteristics by drawing parallels between cancer and infectious diseases. The pervasive fear of contagion in infectious diseases, analogous to misconceptions about cancer, such as the belief in its contagion, highlighted commonalities. Additionally, we found that, like in some infectious diseases, fears about disclosure were a distinct and central component of cancer stigma in this population Through a meticulous exploration, we expanded the framework to encompass dimensions relevant to both infectious and non-infectious disease-related stigma.

Our participants' narratives, encapsulating challenges like financial burdens, fear of death, and recourse to religious and traditional interventions, enriched this exploration. As we revisit the framework, we advocate for its integration with our findings, emphasizing the imperative for enhanced financial support, professional counseling, and community education. This holistic approach recognizes the nuanced interplay between health stigmas in the Tanzanian context, offering valuable insights that transcend the boundaries of specific diseases [13]. Our study contributes to the groundwork for future research and interventions, fostering a comprehensive understanding of the complexities inherent in health-related stigmas.

In this study, we offer insights into the nature of cancer stigma and how it affects treatment and the lives of cancer patients in Tanzania. We hope these findings can guide the development of stigma-reduction interventions, including patient-centered solutions to cancer stigma in the community, in a setting where few studies and no interventions have been carried out on the topic.

We found that in Tanzania, where 49.1% of the population lives below the poverty line [income less than $1.90 per day], financial challenges during cancer treatment can lead to stigma, self-stigmatization, and conflict in families and social relationships. Most healthcare costs, including cancer care, are incurred by the patients, out of pocket at the time-of-service provision, and the high costs of cancer care put an enormous strain on households [26, 27]. Patients often see themselves as a burden to their families and society, and financial challenges can lead to self-stigmatization and serious mental health challenges [28]. Although the relationship between self-stigma and financial challenges has not been extensively studied in Africa, previous research shows that the cost of care is a major concern for cancer patients [29, 30]. Other studies in sub-Saharan countries have also found that a cancer diagnosis can impact a patient's earnings and ability to generate income for the family during treatment [31, 32]. The financial burden of healthcare, also referred to as financial toxicity, is receiving new attention globally [28]. The financial burden of care can also cause patients to seek cheaper alternative treatments, such as traditional or religious healers, or to forego treatment altogether [33]. Furthermore, only approximately 30% of Tanzanians currently have health insurance [34]. Educating the public about health insurance and increasing health insurance coverage could reduce out-of-pocket medical expenses for patients and prevent impoverishment due to cancer treatment.

The country has a population of over 60 million people but fewer than 20 oncologists and only four cancer treatment centers - Aga Khan Hospital in Dar-es-salaam, Bugando

Medical Centre, Kilimanjaro Christian Medical Centre, and the Ocean Road Cancer Institute. This is insufficient for a disease that is one of the leading non-communicable diseases in the country, accounting for 7.6% of deaths [35]. As the country seeks to scale its cancer treatment more broadly, it must dedicate more resources to social services and reduce the emotional burden of care for patients, which has a profound impact on treatment outcomes. The emotional burden of caregiving can be alleviated through the establishment of support groups for both caregivers and patients. These groups serve as safe spaces where individuals can openly share their fears, concerns, and anxieties, with the added benefit of finding practical solutions. Furthermore, resources like counseling provided by mental health professionals should be directed toward cancer patients and their caregivers, with an emphasis on making these services more accessible.

To improve treatment access and reduce their financial stigma, participants suggested providing subsidized or free treatment and increasing access to screening services. The government could play a significant role in reducing barriers to treatment by ensuring sufficient funding and creating public-private or intergovernmental partnerships like the U.S. President's Emergency Plan for AIDS Relief [PEPFAR], which has provided crucial resources in the fight against HIV. Improved technologies such as telehealth can also enable patients to have consultations from wherever they are and only travel to clinics for treatment. Enhancing social support for cancer patients can also positively impact cancer patients' treatment-seeking behavior and overall access to care. This can be achieved through various means, such as establishing peer support groups to provide patients with encouragement and companionship. Furthermore, raising community awareness about the significance of supporting cancer patients through media campaigns or community outreach efforts can help reduce stigma and foster a more supportive environment for these individuals.

This study reveals that cancer patients often face stigma, social isolation, and discrimination. Friends and family members may reject them due to misunderstandings or fears about the disease. However, some patients receive support from their community, reflecting the communal nature of African societies [36]. The fear of social isolation can lead to non-disclosure of the diagnosis and disengagement from care, resulting in emotional distress and seeking alternative treatments [37]. These factors hinder treatment efforts and lead to poor outcomes. Providing social support, including love, companionship, and material/spiritual support, promotes healthy coping strategies and motivates patients to seek treatment despite barriers like stigma and high costs, as observed in previous research on breast cancer in Nigeria [38, 39].

Poor knowledge about cancer contributes to cancer stigma [40]. Our study found no apparent connection between the type or severity of cancer and the response themes. Any cancer diagnosis triggers fear and anxiety due to a general lack of understanding about cancer. Many participants believed education is the best way to combat stigma among patients and the community. Stigmatizing attitudes towards cancer was prevalent and included beliefs that cancer is contagious, always leads to death, and is untreatable, among other myths. Education focused on correcting these misconceptions could increase the readiness and use of cancer screening programs and encourage more patients to seek and adhere to treatment. Additionally, education on the causes of cancer would mitigate self-blame and stigmatizing beliefs that associate cancer with witchcraft and punishment and reduce some of the shame and discomfiture cancer patients experience.

Educational efforts can reduce cancer stigma and social isolation, leading to more social support and better treatment adherence. HIV education campaigns have had some success and could serve as a model for cancer education. Task-shifting to train community health workers to provide counseling support and screening could improve care for cancer patients, as has been done in child healthcare [41, 42] and malarial campaigns [43]. Peer education has

been previously effective in promoting early breast cancer screening in Sub-Saharan Africa [44]. It can be applied to educating and raising awareness about other prevalent cancers, such as prostate and cervical cancer. Technology, such as social media campaigns, could also improve cancer education and awareness.

These findings should be interpreted considering the following limitations. We used convenience sampling to recruit patients capable of completing our survey at a single healthcare site, which excluded patients with very severe symptoms and those who were not attending the clinic. Moreover, future studies may include pediatric patients, family members, and healthcare workers to elicit the perspectives of these groups. The lack of audio recording during interviews may also have resulted in lost content. Despite these limitations, the study's strengths, including the large sample size and representative cross-section of people from diverse geographical locations and the use of Swahili in interviews, enhanced the accuracy and generalizability of the findings to other low-resource regions.

## 5. Conclusions

This research highlights the importance of addressing cancer misconceptions by implementing effective public education campaigns. Moreover, we emphasize the need for the integration of psychosocial services in cancer treatment to provide patients with needed emotional and psychological support. Furthermore, there is a clear need for policy measures to increase access to cancer treatment, particularly in remote and underserved areas, as well as finding innovative ways to reduce the financial burden of cancer for patients and their families. By addressing these issues, we can improve cancer patient care and treatment outcomes.

## Supporting information

**S1 Checklist. Inclusivity in global research.**
(PDF)

## Acknowledgments

The authors would like to express their sincere gratitude to Kilimanjaro Christian Medical Centre in Tanzania for graciously providing the necessary support and facilities for conducting this research. The authors sincerely appreciate the invaluable assistance and welcoming environment the institution provides.

Also, the authors extend their heartfelt appreciation to the participants who generously dedicated their time and actively participated in this study. Without their invaluable contributions, this research would not have been possible.

The authors are also grateful to those who provided guidance, encouragement, and feedback throughout this research endeavor.

## Author Contributions

**Conceptualization:** Judith M. Mwobobia.

**Formal analysis:** Jennifer Headley.

**Funding acquisition:** Brandon A. Knettel.

**Investigation:** Judith M. Mwobobia, Elizabeth F. Msoka, Clotilda S. Tarimo, Victor Katiti.

**Project administration:** Nosayaba Osazuwa-Peters.

**Resources:** Nosayaba Osazuwa-Peters.

**Software:** Erika Juhlin.

**Supervision:** Brandon A. Knettel.

**Writing – original draft:** Judith M. Mwobobia.

**Writing – review & editing:** Brandon A. Knettel.

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
