## [Decision Letter · Decision Letter 0]

17 Aug 2023

PGPH-D-23-01049

“Let Him Die. He Caused It”: A Qualitative Study on Cancer Stigma in Tanzania

Dear Dr. Mwobobia,

Thank you for submitting your manuscript to PLOS Global Public Health. After careful consideration, we feel that it has merit but does not fully meet PLOS Global Public Health’s publication criteria as it currently stands. Therefore, we invite you to submit a revised version of the manuscript that addresses the points raised during the review process.

We look forward to receiving your revised manuscript.

Kind regards,

Rajesh Sharma, Ph.D.

Academic Editor

Journal Requirements:

2. We noticed that you used "unpublished" in the manuscript. We do not allow these references, as the PLOS data access policy requires that all data be either published with the manuscript or made available in a publicly accessible database. Please amend the supplementary material to include the referenced data or remove the references.

3. In the online submission form, you indicated that "The de-identified data for this analysis may be made available upon request with an appropriate data transfer agreement". All PLOS journals now require all data underlying the findings described in their manuscript to be freely available to other researchers, either 1. In a public repository, 2. Within the manuscript itself, or 3. Uploaded as supplementary information.

Additional Editor Comments (if provided):

Please revise the manuscript in the light of reviewer's comments and suggestions. Please make sure the manuscript is proofread by native English speaker before re-submission to the journal.

Reviewers' comments:

Reviewer's Responses to Questions

**Comments to the Author**

1. Does this manuscript meet PLOS Global Public Health’s publication criteria? Is the manuscript technically sound, and do the data support the conclusions? The manuscript must describe methodologically and ethically rigorous research with conclusions that are appropriately drawn based on the data presented.

Reviewer #1: Yes

Reviewer #2: Partly

Reviewer #3: Yes

2. Has the statistical analysis been performed appropriately and rigorously?

Reviewer #1: Yes

Reviewer #2: Yes

Reviewer #3: Yes

3. Have the authors made all data underlying the findings in their manuscript fully available (please refer to the Data Availability Statement at the start of the manuscript PDF file)?

Reviewer #1: Yes

Reviewer #2: Yes

Reviewer #3: Yes

4. Is the manuscript presented in an intelligible fashion and written in standard English?

Reviewer #1: Yes

Reviewer #2: Yes

Reviewer #3: Yes

5. Review Comments to the Author

Reviewer #1: The review is well written with obviously no mistakes. There is some mistyping in the references section: reference # 5 starts with (36.), which should be deleted; reference number 22 should specify the version used; and references in page 35, the font was changed.

Reviewer #2: The manuscript fails to meet:

1. The submission guideline of the journal

2. Establishing knowledge gap

3. Appropriate study design and its details including the type of study design for instance in the abstract section.

4.Data quality assurance

6. Grammer, language and statistics

7. Inconsistent from the beginning to the end of the manuscript

Reviewer #3: Great introduction

Study design:

- Because the study focuses on stigma, I would want to know how participants were approached during recruitment and if selected for a specific subset of participants.

- Was monetary motivation the principal factor for participants to join the research?

Figure 1 and 2 are missing

Results:

"Th responses of the study participants" and a number of other spelling errors

In the "Beliefs Contributing to Cancer Stigma" in the "Death sentence, associated with

death" section, a different quote than the one above would make a stronger point, especially if this sentiment is found with 78% frequency. Repeating quotes is useful but can make a stronger point by alluding to the quote already presented and providing a different participant quote illustrating the same theme.

"Negative body image and self hate" the quote shared talks specifically of skin cancer, does this theme extend beyond skin cancer? and if yes, how it is represented?

Did the research try to understand where these notions of stigma stem from? Ex: why do community members believe cancer to be contagious? This could be an interesting discussion to really get at different solutions (besides education) to advancing cancer care and diminishing stigma.

How do the author propose the emotional burden of care for patients can be reduced? Similarly, how can social support for cancer patients be promoted to increase access to care?

6. PLOS authors have the option to publish the peer review history of their article (what does this mean?). If published, this will include your full peer review and any attached files.

**Do you want your identity to be public for this peer review?** For information about this choice, including consent withdrawal, please see our Privacy Policy.

Reviewer #1: **Yes: **Inas Saleh Almazari

Reviewer #2: No

Reviewer #3: No

---

## [Decision Letter · Decision Letter 1]

11 Dec 2023

PGPH-D-23-01049R1

“Let Him Die. He Caused It”: A Qualitative Study on Cancer Stigma in Tanzania

Dear Dr. Mwobobia,

Thank you for submitting your manuscript to PLOS Global Public Health. After careful consideration, we feel that it has merit but does not fully meet PLOS Global Public Health’s publication criteria as it currently stands. Therefore, we invite you to submit a revised version of the manuscript that addresses the points raised during the review process.

Two of the previous three reviewers have re-evaluated the revision. One reviewer still has some concerns.

We look forward to receiving your revised manuscript.

Kind regards,

Jianhong Zhou

Staff Editor

Journal Requirements:

Additional Editor Comments (if provided):

Reviewers' comments:

Reviewer's Responses to Questions

**Comments to the Author**

1. If the authors have adequately addressed your comments raised in a previous round of review and you feel that this manuscript is now acceptable for publication, you may indicate that here to bypass the “Comments to the Author” section, enter your conflict of interest statement in the “Confidential to Editor” section, and submit your "Accept" recommendation.

Reviewer #1: All comments have been addressed

Reviewer #2: All comments have been addressed

2. Does this manuscript meet PLOS Global Public Health’s publication criteria? Is the manuscript technically sound, and do the data support the conclusions? The manuscript must describe methodologically and ethically rigorous research with conclusions that are appropriately drawn based on the data presented.

Reviewer #1: Yes

Reviewer #2: Partly

3. Has the statistical analysis been performed appropriately and rigorously?

Reviewer #1: Yes

Reviewer #2: (No Response)

4. Have the authors made all data underlying the findings in their manuscript fully available (please refer to the Data Availability Statement at the start of the manuscript PDF file)?

Reviewer #1: Yes

Reviewer #2: Yes

5. Is the manuscript presented in an intelligible fashion and written in standard English?

Reviewer #1: Yes

Reviewer #2: Yes

6. Review Comments to the Author

Reviewer #1: The previous comments were reviewed and made accurately.

Reviewer #2: We acknowledge the authirs for conducting such interesting study as a team and the following are our main concerns ;

1.The scope of the issue is inconsistently addressed throughout the document.

2.Logical flow of the manuscript is lacking e.g. First stigma.and then its outcome i.e. Descriminatiin and its tyoes.

3. The type of the qualitative research is not described.

4. The steps taken in the analysis were missed.

5. The role of the third and the consequent authors and the one stated in the methods section contraindicts. Furthermore , the role of authors is lacking .

6. Most of the study partipants are at stage IV.Hence, how did you manage social desirablity bias.

7. Lacks logical flow and especially the results section only presents the age and the tyoe of cancer, why?

8. Again, the results section took wider space than its fair share.

9. How we compare a one day experience with 180 days experience ?

10. Maintain consistency, clarity and ensure whether the contents of all the sub sections contains all the contents

11. The data quality assurance methods are incomplete.

12. How can we extrapolate the Infectious disease related stigma with non infectious one and yry to meticulously revisit the framework .

Regards,

7. PLOS authors have the option to publish the peer review history of their article (what does this mean?). If published, this will include your full peer review and any attached files.

**Do you want your identity to be public for this peer review?** For information about this choice, including consent withdrawal, please see our Privacy Policy.

Reviewer #1: No

Reviewer #2: No

---

## [Decision Letter · Decision Letter 2]

4 Mar 2024

PGPH-D-23-01049R2

“Let Him Die. He Caused It”: A Qualitative Study on Cancer Stigma in Tanzania

Dear Dr. Mwobobia,

Thank you for submitting your manuscript to PLOS Global Public Health. After careful consideration, we feel that it has merit but does not fully meet PLOS Global Public Health’s publication criteria as it currently stands. Therefore, we invite you to submit a revised version of the manuscript that addresses the points raised during the review process.

The reviewer raised new concerns for this revision. Please see the comments below.

We look forward to receiving your revised manuscript.

Kind regards,

Jianhong Zhou

Staff Editor

Journal Requirements:

1. Please send a completed 'Competing Interests' statement, including any COIs declared by your co-authors. If you have no competing interests to declare, please state "The authors have declared that no competing interests exist". Otherwise please declare all competing interests beginning with the statement "I have read the journal's policy and the authors of this manuscript have the following competing interests:"

2. We have noticed that you have uploaded Supporting Information files, but you have not included a list of legends. Please add a full list of legends for your Supporting Information files after the references list.

Additional Editor Comments (if provided):

Reviewers' comments:

Reviewer's Responses to Questions

**Comments to the Author**

1. If the authors have adequately addressed your comments raised in a previous round of review and you feel that this manuscript is now acceptable for publication, you may indicate that here to bypass the “Comments to the Author” section, enter your conflict of interest statement in the “Confidential to Editor” section, and submit your "Accept" recommendation.

Reviewer #2: All comments have been addressed

2. Does this manuscript meet PLOS Global Public Health’s publication criteria? Is the manuscript technically sound, and do the data support the conclusions? The manuscript must describe methodologically and ethically rigorous research with conclusions that are appropriately drawn based on the data presented.

Reviewer #2: Partly

3. Has the statistical analysis been performed appropriately and rigorously?

Reviewer #2: Yes

4. Have the authors made all data underlying the findings in their manuscript fully available (please refer to the Data Availability Statement at the start of the manuscript PDF file)?

Reviewer #2: Yes

5. Is the manuscript presented in an intelligible fashion and written in standard English?

Reviewer #2: Yes

6. Review Comments to the Author

Reviewer #2: We thank the authors for revising the manuscript based on the comments given in the previous version. But the point to point response is not sent to us. The following are my comments in this version;

1. The scope and the inner content of the article have inconsistency.

2. Is the main objectibe of the study stigma or descrimination? If so are you disicussing the ocvukt or overt character?ideal or practical?

3.The purpose of the article is stigma in the health care settings. However , it also includes individual, family and societal attitudes regarding the cancer patients among the populations. Hence I recommend you to make it one.

4. The comments in the methods section is not well addressed. E.g. Data quality assurance and lacks reference for example Guba 1972. Hence, the methods section still needs major revision.

5. The philosophic stance of the researcher is out of the philosophy of qualitative study.

-The type of qualitative study

-The duration of the interview etc

6. The resukt, dicussion and the consequent section needs major and critical revision. In addition , shoukd be short and precise .

7. Reference needs revisit.

Regards,

7. PLOS authors have the option to publish the peer review history of their article (what does this mean?). If published, this will include your full peer review and any attached files.

**Do you want your identity to be public for this peer review?** For information about this choice, including consent withdrawal, please see our Privacy Policy.

Reviewer #2: No

---

## [Decision Letter · Decision Letter 3]

7 May 2024

“Let Him Die. He Caused It”: A Qualitative Study on Cancer Stigma in Tanzania

PGPH-D-23-01049R3

Dear Ms Mwobobia,

We are pleased to inform you that your manuscript '“Let Him Die. He Caused It”: A Qualitative Study on Cancer Stigma in Tanzania' has been provisionally accepted for publication in PLOS Global Public Health.

Please note you will see additional reviewer comments below; it is not necessary to revise your paper further nor respond to the comments. They are provided for your interest only.

Best regards,

Julia Robinson

Staff Editor

Reviewer Comments (if any, and for reference):

Reviewer's Responses to Questions

**Comments to the Author**

1. If the authors have adequately addressed your comments raised in a previous round of review and you feel that this manuscript is now acceptable for publication, you may indicate that here to bypass the “Comments to the Author” section, enter your conflict of interest statement in the “Confidential to Editor” section, and submit your "Accept" recommendation.

Reviewer #2: All comments have been addressed

2. Does this manuscript meet PLOS Global Public Health’s publication criteria? Is the manuscript technically sound, and do the data support the conclusions? The manuscript must describe methodologically and ethically rigorous research with conclusions that are appropriately drawn based on the data presented.

Reviewer #2: Yes

3. Has the statistical analysis been performed appropriately and rigorously?

Reviewer #2: N/A

4. Have the authors made all data underlying the findings in their manuscript fully available (please refer to the Data Availability Statement at the start of the manuscript PDF file)?

Reviewer #2: Yes

5. Is the manuscript presented in an intelligible fashion and written in standard English?

Reviewer #2: Yes

6. Review Comments to the Author

Reviewer #2: We appreciate the authors for addressing the comments given in the previous version and the following are my comments in this version;

1. The scope of the study and the title has little inconsistency e.g. "Let him die " is more of practilical.

2. The abstract is incomplete e.g. study design.

3. The methods lack Brief elaboration of the setting with references, the sampling methods, the socio economic settings, thetrust worthiness by citing appropriate references.

4. The themes and the sub themes and how to deal with it in the results and the consequent section needs high concern

5. High sample in one settings needs also further elaboration and think whether you have dealt with discrimination or stigma related to cancer and the theoretical framework used as well as the philosophical stance of your study.

Regards,

7. PLOS authors have the option to publish the peer review history of their article (what does this mean?). If published, this will include your full peer review and any attached files.

**Do you want your identity to be public for this peer review?** For information about this choice, including consent withdrawal, please see our Privacy Policy.

Reviewer #2: No
